# Accelerated pseudogenization on the neo-X chromosome in *Drosophila miranda*

Masafumi Nozawa[1,2,3], Kanako Onizuka[1], Mai Fujimi[1], Kazuho Ikeo[1,2] & Takashi Gojobori[1,4]

Y chromosomes often degenerate via the accumulation of pseudogenes and transposable elements. By contrast, little is known about X-chromosome degeneration. Here we compare the pseudogenization process between genes on the neo-sex chromosomes in *Drosophila miranda* and their autosomal orthologues in closely related species. The pseudogenization rate on the neo-X is much lower than the rate on the neo-Y, but appears to be higher than the rate on the orthologous autosome in *D. pseudoobscura*. Genes under less functional constraint and/or genes with male-biased expression tend to become pseudogenes on the neo-X, indicating the accumulation of slightly deleterious mutations and the feminization of the neo-X. We also find a weak trend that the genes with female-benefit/male-detriment effects identified in *D. melanogaster* are pseudogenized on the neo-X, implying the masculinization of the neo-X. These observations suggest that both X and Y chromosomes can degenerate due to a complex suite of evolutionary forces.

[1] Center for Information Biology, National Institute of Genetics, 1111 Yata, Mishima, Shizuoka 411-8540, Japan. [2] Department of Genetics, SOKENDAI, 1111 Yata, Mishima, Shizuoka 411-8540, Japan. [3] Department of Biological Sciences, Tokyo Metropolitan University, 1-1 Minamiosawa, Hachioji, Tokyo 192-0397, Japan. [4] King Abdullah University of Science and Technology (KAUST), Computational Bioscience Research Center, Biological and Environmental Science and Engineering, 4700 KAUST, Thuwal 23955-6900; Kingdom of Saudi Arabia. Correspondence and requests for materials should be addressed to M.N. (email: manozawa@tmu.ac.jp).

Sex chromosomes have repeatedly emerged in many different lineages during the long history of life[1]. A well-known phenomenon after the emergence of sex chromosomes is degeneration of a Y (or W) chromosome[2–5]. Although there are cases in which the Y (or W) chromosome has not degenerated very much[1], the Y chromosome has indeed degenerated greatly in a variety of lineages, including many of mammals[2], birds[6], snakes[7] and insects[8]. Degeneration of the Y is largely explained by recombination suppression between the X and Y chromosomes and a reduction in population size, both of which make natural selection inefficient[9,10]. Consequently, pseudogenes and transposable elements accumulate on the Y.

Compared with this well-known trajectory of the Y chromosome, comparatively little is known about the evolution of X chromosomes after their emergence. In particular, to the best of our knowledge, no one has empirically examined the possibility of X-chromosome degeneration. Although the degeneration process of the X as well as of the Y is essential for understanding the evolution of sex chromosomes after their emergence, it has been, in general, very difficult to trace the steps of degeneration evolutionarily. This is because the sex chromosomes are ancient in many species, including in model organisms such as humans, mice and *Drosophila melanogaster*, which has hindered reconstruction of the ancestral states of sex chromosomes at their emergence. Consequently, studies of these organisms have not been able to reveal whether the X chromosome degenerates or not.

In this study, we utilized *D. miranda*, which has experienced two recent chromosomal fusions between sex chromosomes and ordinary autosomes, resulting in three sex chromosomes with different ages[11]. Among them, the youngest sex chromosomes, the so-called neo-sex chromosomes, emerged only ∼1 million years ago (Mya) in the *D. miranda* lineage after splitting from *D. pseudoobscura*[12]. Since neo-sex chromosomes are in an early stage of sex chromosome evolution[3], they are ideal materials to trace pseudogenization events on the neo-X and neo-Y compared with an orthologous autosome in the closely related species *D. pseudoobscura*, with the autosome in *D. obscura* as an outgroup. We therefore massively sequenced the genomes and transcriptomes of these three species with the utilization of the available high-quality genome sequences of *D. pseudoobscura*[13] and *D. miranda*[3,14], investigating the process of pseudogenization of the neo-X chromosome. We here report that the rate of pseudogenization on the neo-X is indeed higher than that on autosomes due to a complex suite of evolutionary forces including feminization and possibly masculinization.

## Results

**Higher pseudogenization rate on the neo-X than on autosomes.** To investigate the detailed process of pseudogenization of the neo-sex chromosomes in *D. miranda*, we focused on the three species, *D. miranda*, *D. pseudoobscura* and *D. obscura*. *Drosophila pseudoobscura* is one of the closest relatives of *D. miranda*; it diverged ∼2 Mya and does not have the neo-sex chromosomes[12]. *Drosophila obscura* is one of the closest relatives with the ancestral *Drosophila* sex chromosome system (Fig. 1). We first conducted a *de novo* sequencing of the *D. obscura* genome and resequencing of the *D. miranda* genome. In addition, we performed extensive RNA sequencing (RNA-seq) to construct transcriptomes of these three species (Supplementary Tables 1–7, see also Methods and Supplementary Methods for details).

Our annotation based on these sequence data showed that nearly two-thirds of the neo-Y genes in *D. miranda* are already pseudogenized (either silenced, disrupted or both; Fig. 2a), roughly consistent with previous observations[3,15]. The proportion

of pseudogenes on the neo-X was much lower than that on the neo-Y, but approximately one-third of genes on the neo-X were pseudogenes (Fig. 2a). This observation raised the possibility that the pseudogenization rate accelerates after a chromosome becomes an X chromosome. Accordingly, we analysed *D. miranda* and *D. pseudoobscura* genes on the Muller element C (the neo-sex chromosomes and chromosome 3 in *D. miranda* and *D. pseudoobscura*, respectively), the orthologues on which are likely to be functional in *D. obscura* and not to have experienced any inter-chromosomal translocation after the species split from *D. obscura*. Using these 1,282 orthologues and a parsimony framework, we assigned pseudogenization events to each of the lineages after divergence from *D. obscura* (see Methods and Supplementary Methods for detailed procedures). As expected, the rate of pseudogenization was much lower on the neo-X than on the neo-Y (18.9% versus 58.4%, respectively; $\chi^2$ test: $\chi^2 = 259.4$, d.f. = 1, $P = 2.3 \times 10^{-58}$; Fig. 2b). However, we were interested in knowing whether the pseudogenization rate is higher on the neo-X than on autosomes. Since *D. miranda* diverged from *D. pseudoobscura* ∼2 Mya and the neo-sex chromosomes emerged ∼1 Mya[12], we divided the number of pseudogenization events (229) on the *D. pseudoobscura* lineage in half (114.5 for each of the two), assuming that the rate of pseudogenization in the *D. pseudoobscura* lineage is constant. The pseudogenization rate of the neo-X was about twofold compared with that of its orthologous autosome (chromosome 3) in *D. pseudoobscura* (18.9% versus 8.9%; $\chi^2$ test: $\chi^2 = 45.2$, d.f. = 1, $P = 1.8 \times 10^{-11}$; Fig. 2b). This difference in pseudogenization rate is unlikely to be explained by interspecific differences because other homologous chromosomes (so-called Muller elements) did not show conspicuous differences in pseudogenization rates between species (Supplementary Fig. 1). Accelerated pseudogenization on the neo-X was also supported by another comparison in which the pseudogenization rate after becoming a neo-X was more than twofold compared with that before becoming a neo-X (18.9% versus 6.6%; $\chi^2$ test: $\chi^2 = 76.6$, d.f. = 1, $P = 2.1 \times 10^{-18}$; Fig. 2b). Therefore, the difference in the pseudogenization rate on this chromosome (the neo-X in *D. miranda* and chromosome 3 in *D. pseudoobscura*) is most likely due to sex linkage.

The above analyses depend on the times when *D. miranda* and *D. pseudoobscura* diverged and when the neo-sex chromosomes emerged. Since the neo-sex chromosomes could have emerged 1.5 Mya[16], we also conducted the analysis based on this emergence time. The result showed that the difference in the pseudogenization rate between the neo-X and its orthologous autosome (chromosome 3) in *D. pseudoobscura* remained significant ($\chi^2$ test: $\chi^2 = 11.8$, d.f. = 1, $P = 5.8 \times 10^{-4}$; Supplementary Fig. 2). It should also be mentioned that our definition of disrupted genes [that is, less than 80% in the coding sequence (CDS) length compared with the average of other orthologous members, see Methods and Supplementary Methods for details] was set operationally. Therefore, we also examined other thresholds to further evaluate the possibility of accelerated pseudogenization on the neo-X. The number of pseudogenization events on each lineage varied to some extent depending on the threshold used (Supplementary Table 8). However, the result consistently supported that the rate of pseudogenization on the neo-X lineage is faster than the rate on the orthologous autosome in *D. pseudoobscura* (see X versus Pse in Supplementary Table 9).

Another potential concern is the method of gene annotation. While both female and male RNA-seq reads were utilized for reconstructing transcripts on autosomes, only female reads were used for reconstructing the neo-X transcripts. This approach was necessary to clearly separate the reads derived from the neo-X

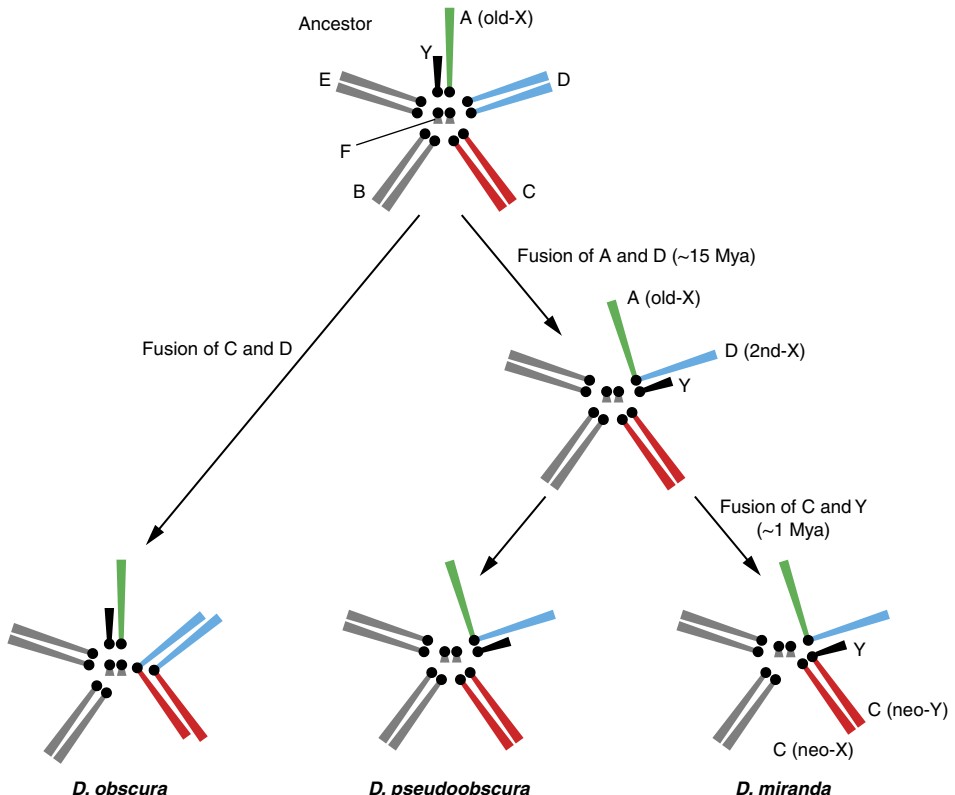

**Figure 1 | Evolution of karyotypes in *Drosophila miranda* and its closely related species.** Male karyotypes are shown. In *D. obscura* (as well as in the common ancestor of these three species), only the Muller element A is the X, whereas in *D. pseudoobscura* the Muller element D is also X (2nd-X or XR). In *D. miranda*, the Muller element C additionally became the neo-sex chromosomes. More specifically, one of the pairs was fused with Y, and the other consequently became the neo-X. In other words, *D. miranda* has three X chromosomal arms with different ages. Grey bars represent autosomes, whereas green bars are the old-X. Blue bars indicate the 2nd-X or its orthologous autosomes. Red bars represent the neo-X, neo-Y, or its orthologous autosomes. Black bars are Y chromosomes. Black dots indicate the rough positions of centromere.

and those from the neo-Y (see Methods and Supplementary Methods for details). However, if a neo-X-linked gene is exclusively expressed in males, the gene may falsely be classified as silenced pseudogenes in our analysis, possibly resulting in an overestimation of the number of pseudogenization events on the neo-X lineage. To solve this issue, we also used the genes that were annotated without using RNA-seq data but just based on the genome sequence for examining gene expression. In this approach, even if a gene was predicted based on a genome sequence, the gene was regarded to be functional if its mRNA was detected in a tissue with a FPKM (fragments per kilobase of exon per million mapped reads) value of $\geq 1$. This treatment should remove the bias of the gene annotation on the neo-X, because by doing so the genes that are only expressed in males can correctly be classified as functional genes. In other words, this analysis would theoretically reduce the number of pseudogenes (or silenced genes in a strict sense) on the neo-X and possibly the number on autosomes as well. Indeed, the number of pseudogenization events became smaller on all lineages (Supplementary Fig. 3). Yet, the number on the neo-X lineage remained significantly greater than the number on the corresponding autosomal lineage in *D. pseudoobscura* (*X* versus *Pse* in Supplementary Fig. 3). Therefore, these results consistently indicate that not only becoming a Y chromosome, but also becoming an X chromosome, accelerates the rate of pseudo-genization in their early stage of evolution after emergence.

According to Zhou and Bachtrog[17], most of the loss-of-function mutations on the neo-Y in *D. albomicans*, another species with independent neo-sex chromosomes, have been initiated by transcriptional silencing, rather than by the loss of protein-coding potential. We also found that pseudogenization caused by silencing or decay in regulatory functions dominates the early stage of the neo-Y degeneration in *D. miranda* (Fig. 2a). On the neo-X, however, pseudogenization caused by the loss of protein-coding potential was more common than pseudogenization caused by regulatory mutations (Fig. 2a). Therefore, the initial steps in pseudogenization might be quite different between the X and Y chromosomes. This difference between the neo-X and the neo-Y may be explained by meiotic sex chromosome inactivation (MSCI; gene silencing of sex chromosomes in the heterogametic sex during meiosis), which is believed to operate in many animals to prevent meiotic drivers from invading sex chromosomes that could be particularly fragile in the heterogametic sex[18]. Therefore, the degeneration of the neo-Y (but not much of the neo-X) might be initiated by gene silencing through MSCI[19], although the connection between MSCI and the neo-Y silencing at the molecular level remains to be clarified. Note, however, that our annotation may have potentially overestimated the number of silenced pseudogenes as described above. Indeed, when we also used the annotated genes based on the genome sequence for gene expression as above, a majority of genes were regarded to be expressed genes although many of them were expressed very weakly only in a particular tissue (that is, FPKM values of just above 1). Under this condition, the number of silenced pseudogenes became smaller than the number of disrupted pseudogenes on both neo-X and neo-Y (Supplementary Fig. 4). Therefore, it will be difficult to reach any decisive conclusion regarding the causal mutations of

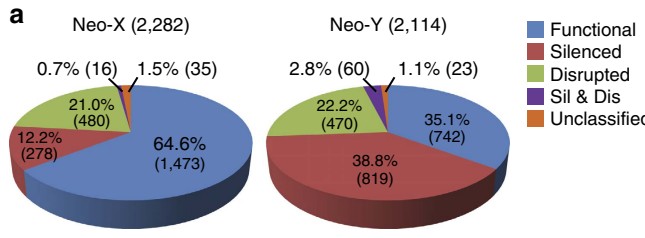

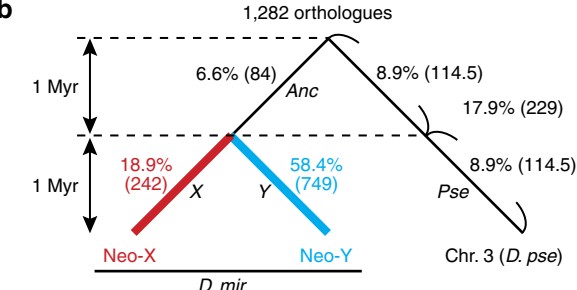

**Figure 2 | Pseudogenization on the neo-X and neo-Y. (a)** Classification of neo-X and neo-Y genes. The numbers in parentheses are the numbers of genes in each category. Blue, red, green, purple and orange colours mean functional, silenced, disrupted, silenced-and-disrupted, and unclassified genes, respectively. Silenced, disrupted, and silenced-and-disrupted genes were regarded as pseudogenes. **(b)** Proportions of genes that became pseudogenized in each evolutionary lineage. The 1,282 orthologues that are likely to be functional in *D. obscura* and not to have experienced any inter-chromosomal translocation after the species split from *D. obscura* were analysed. The *D. pseudoobscura* autosome that is orthologous to the neo-X and neo-Y is chromosome 3 (Muller element C). Pseudogenization events in the lineage leading to *D. pseudoobscura* were equally split in half assuming 2 Mya for the divergence time of *D. miranda-D. pseudoobscura*, 1 Mya for the emergence time of the neo-sex chromosomes, and a constant rate of pseudogenization in the *D. pseudoobscura* lineage. Rate of pseudogenization: branch $X$ < branch $Y$ ($\chi^2$ test: $\chi^2 = 259.4$, d.f. = 1, $P = 2.3 \times 10^{-58}$), branch $X$ > branch $Pse$ ($\chi^2$ test: $\chi^2 = 45.2$, d.f. = 1, $P = 1.8 \times 10^{-11}$), and branch $X$ > branch $Anc$ ($\chi^2$ test: $\chi^2 = 76.6$, d.f. = 1, $P = 2.1 \times 10^{-18}$). Branch names are as follows: *Anc*, the ancestral branch before separating the neo-X and neo-Y; *X*, the neo-X branch; *Y*, the neo-Y branch; and *Pse*, the *D. pseudoobscura* branch after splitting from *D. miranda*. The number of orthologues that were used for the analysis with functional genes in *D. obscura* is shown above the tree. The numbers in parentheses are the numbers of genes that were pseudogenized in each lineage. Results for other chromosomes are shown in Supplementary Fig. 1. Results based on different conditions are shown in Supplementary Figs 2, 3 and 4 and in Supplementary Tables 8 and 9.

pseudogenization on the neo-sex chromosomes. Further studies on different young sex chromosomes are necessary to determine whether these patterns of degeneration of the X and Y are robust and generalized.

What kinds of evolutionary forces have influenced the rate of pseudogenization on the neo-X? Theoretically, if recombination suppression occurs between the X and Y chromosomes, natural selection becomes less efficient not only on the Y but also on the X, because the X cannot remove deleterious mutations through recombination in males. In addition, the effective population size of the X chromosomes is expected to be three-fourths compared with the size of an autosome, because a male has only one X chromosome[20]. Therefore, the X chromosome could also degenerate by the same mechanisms that cause Y degeneration, but to a lesser extent. Contrary to the above statement, however, such reduction in efficacy of natural selection may be unlikely to

cause the X-chromosome degeneration in *Drosophila* species, because meiotic recombination is restricted to females in many *Drosophila* species[21,22]. Therefore, the proportion of recombining chromosomes for the X is two-thirds, whereas the proportion for autosomes is two-fourths[23]. In this case, unlike in other organisms, natural selection is expected to be four-thirds times more efficient on the X than on autosomes, which offsets a reduction in the effective population size of the X chromosomes by three-fourths compared with autosomes. Although other factors such as the effective numbers of breeding males and females may also affect the efficacy of natural selection on different chromosomes[24], this theoretical prediction suggests that the efficacy of natural selection on the X and autosomes could be similar in many *Drosophila* species[25]. In the following, we therefore considered other possibilities for this accelerated pseudogenization. However, we do not necessarily exclude the possibility of reduction in the efficacy of natural selection to explain the accelerated pseudogenization on the neo-X, as discussed in the next section.

**Pseudogenization of genes under less functional constraint.** We first asked what kinds of genes tend to remain functional or become pseudogenized on the neo-X. Based on a gene ontology analysis[26], we found that genes involved in developmental and cellular processes as well as genes that function in organelles tend to be functional on the neo-X (Supplementary Table 10).

Next, we examined the ratio of non-synonymous to synonymous nucleotide distance (the $d_N/d_S$ ratio). For this analysis, we used the same 1,282 orthologues that are likely to be functional in *D. obscura* as above to make sure that the genes were functional in the common ancestor of these three species. All of these orthologues were classified based on the functionality on the neo-X and neo-Y, that is, $X_F$-$Y_F$ (functional on both neo-X and neo-Y), $X_F$-$Y_P$ (functional on the neo-X and non-functional on the neo-Y), $X_P$-$Y_F$ (non-functional on the neo-X and functional on the neo-Y), and $X_P$-$Y_P$ (non-functional on both neo-X and neo-Y). On the branch $X$, we observed a significantly higher $d_N/d_S$ ratio for genes that are pseudogenized on the neo-X ($X_P$-$Y_F$ and $X_P$-$Y_P$) than for genes that are functional on the neo-X ($X_F$-$Y_F$ and $X_F$-$Y_P$) (Fig. 3b). Of course, this difference can be explained, at least partly, by the loss of functional constraints after a functional gene became a pseudogene on the neo-X. However, the ancestral genes from which these pseudogenes are derived also showed a similar trend particularly for the genes belonging to the $X_P$-$Y_F$ category (Fig. 3a). Therefore, pseudogenes on the neo-X were under less functional constraint when they were functional and located on the autosome. In other words, less important genes tended to have been pseudogenized after the origin of the neo-X. Therefore, reduction in the efficacy of natural selection on the neo-X could still be a part of the reasons for the accelerated pseudogenization on the neo-X. Note also that some of the $X_P$-$Y_F$ genes that show high $d_N/d_S$ ratios (in particular greater than 1) could be under sexual selection and/or function exclusively in testis or accessory gland, because those genes are known to evolve fast[27,28]. This speculation is indeed consistent with the analyses in the following sections.

Neo-Y-linked pseudogenes ($X_F$-$Y_P$ and $X_P$-$Y_P$) also showed higher $d_N/d_S$ ratios than the neo-Y-linked functional genes ($X_F$-$Y_F$ and $X_P$-$Y_F$) (Fig. 3c). However, the ancestral lineage (branch *Anc*) did not show any such pattern (Fig. 3a), indicating that the elevated $d_N/d_S$ ratio on the neo-Y pseudogenes is a consequence of pseudogenization after losing their functions. In other words, functional constraints on genes before becoming the neo-Y do not really affect the fates of the genes after becoming

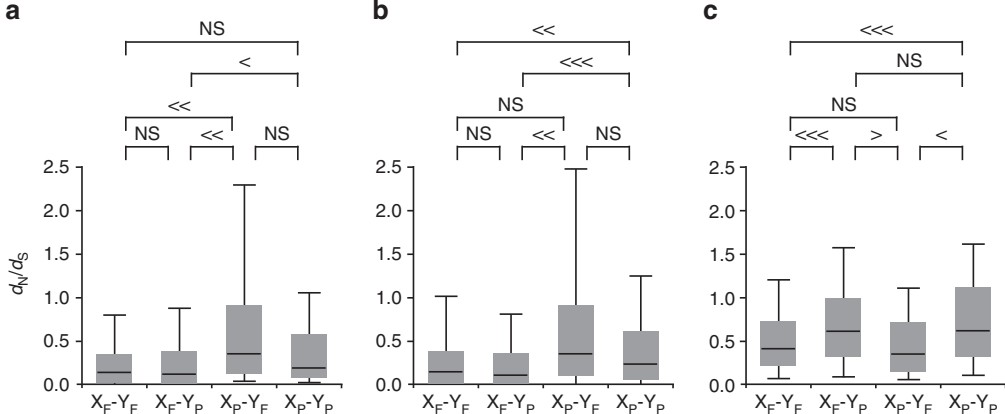

**Figure 3 | Ratios of non-synonymous to synonymous nucleotide distance.** The 1,282 orthologues that are likely to be functional in *D. obscura* and not to have experienced any inter-chromosomal translocation after the species split from *D. obscura* were analysed. The lines in the boxes represent medians, 50% of values are included in the boxes, and 80% of values are included within the bars. $X_F$-$Y_F$, a group of genes whose orthologues are functional on both neo-X and neo-Y; $X_F$-$Y_P$, a group of genes whose orthologues are functional on the neo-X and non-functional on the neo-Y; $X_P$-$Y_F$, a group of genes whose orthologues are non-functional on the neo-X and functional on the neo-Y; $X_P$-$Y_P$, a group of genes whose orthologues are non-functional on both neo-X and neo-Y. Statistical significance was calculated based on a Monte Carlo simulation with 1,000 bootstrap replicates: $\gg\gg$ or $\lll$, $P<0.001$; $\gg$ or $\ll$, $P<0.01$; $>$ or $<$, $P<0.05$; NS, $P\geq0.05$. Note that the Mann-Whitney $U$ test also showed the same trend. Ratios of non-synonymous to synonymous nucleotide distance ($d_N/d_S$ ratios) on (**a**) the ancestral branch before separating the neo-X and neo-Y, (**b**) the neo-X branch, and (**c**) the neo-Y branch.

the neo-Y. Therefore, patterns of pseudogenization are quite different between the neo-X and neo-Y.

**Accelerated pseudogenization by feminization.** Sex chromosomes show biased transmission. More specifically, two-thirds of X chromosomes are inherited through females, while Y chromosomes are inherited only through males. This biased transmission may also affect pseudogenization on the neo-X. We hypothesized that genes that are unimportant for females tend to be pseudogenized on the neo-X because of this biased transmission. In the hypothetical case of a gene located on a pair of autosomes having a function essential in males, but not in females, pseudogenization would not be expected as long as the gene is located on an autosome. However, if the pair of autosomes become sex chromosomes, the X-linked gene could be released from functional constraint to some extent and eventually undergo pseudogenization; the deleterious effect of the loss of the X-linked gene in males could be minimized if the Y-linked homologue remains functional. Under the assumption that genes with male-biased expression are less important for females, we expected the $X_P$-$Y_F$ genes to show male-biased expression in the ancestor, in which the neo-X as well as the neo-Y were still a pair of autosomes. However, since we cannot examine the gene expression level in the ancestor, we used a closely related species, *D. pseudoobscura*, as a proxy for the ancestral species. Our hypothesis predicts that the female to male expression ratio (F/M ratio) in *D. pseudoobscura* is particularly low for genes whose orthologues are pseudogenized on the neo-X but functional on the neo-Y compared with other categories of genes. For this analysis, we examined 1,672 orthologues that are expressed on the Muller element C (chromosome 3) in *D. pseudoobscura*.

We observed that the $X_P$-$Y_F$ genes whose orthologues are pseudogenized on the neo-X but not on the neo-Y indeed showed the lowest F/M ratio (Fig. 4a–c). In particular, we observed a conspicuous difference between the $X_P$-$Y_F$ genes and other three categories of genes when comparing the expression of these genes in ovaries and testes (Fig. 4a, see also Supplementary Fig. 5 for other tissues). By contrast, we detected a smaller difference when we removed gonad tissues from the abdomens (Fig. 4c; see also Supplementary Fig. 5 for other somatic tissues). In general, the

gonad-containing tissues tend to show large differences between the $X_P$-$Y_F$ genes and other three categories of genes, whereas the tissues without gonads showed small differences (Supplementary Fig. 5). This trend is consistent with our expectations because the functions of genes expressed in testis (or other male gonads) are unlikely to be important for females and are thus pseudogenized on the neo-X. Indeed, we found that genes that showed the highest expression in testis (or accessory gland to a lesser extent) tended to be pseudogenized on the neo-X (Fig. 4d), consistent with the previous study[16]. By contrast, the proportion of pseudogenes was relatively low for genes with the highest expression in ovary. It should be noted that this trend is specific for the neo-X but not for other chromosomes (Supplementary Fig. 6), indicating that these pseudogenization events are largely due to the linkage to the neo-X. One may suspect that our finding is an artefact due to our approach of gene annotation and/or classification. More specifically, if a neo-X-linked gene is exclusively expressed in males (that is, a low F/M ratio), our approach may falsely judge a gene as a silenced pseudogene as mentioned repeatedly. In this case, our finding may be an artefact, because in such case it would be natural that the orthologue in *D. pseudoobscura* also shows a low F/M ratio. To evaluate this possibility, we again adopted the approach in which the genes predicted solely on a genome can be regarded as functional genes if their hypothetical transcripts are detected in the RNA-seq data. However, the trend of our original observations remained to be essentially the same, although the difference between the $X_P$-$Y_F$ genes and other categories of genes became smaller (Supplementary Fig. 7). Therefore, our finding would not be an artefact but reflect a nature. It should also be mentioned that the difference between $X_P$-$Y_F$ and $X_P$-$Y_P$ genes became insignificant in the reanalysis, which may be reasonable if orthologues of the $X_P$-$Y_P$ genes are pseudogenized in *D. pseudoobscura* but still expressed in male-specific tissues particularly in testis, where many genes are expressed due to the transcriptionally active chromatin state[29]. Therefore, our conservative re-analysis also supports the idea that a certain proportion of male-biased genes were pseudogenized on the neo-X. These results collectively indicate that pseudogenization on the neo-X was accelerated by feminization (or demasculinization in a more strict sense).

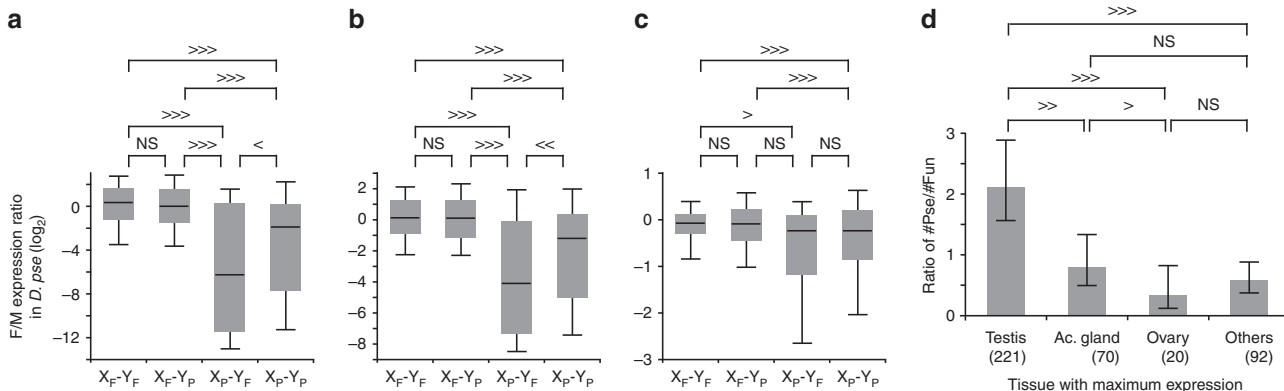

**Figure 4 | Relationship between pseudogenization and gene expression.** Statistical significance was calculated based on a Monte Carlo simulation with 1,000 bootstrap replicates: $\ggg$ or $\lll$, $P < 0.001$; $\gg$ or $\ll$, $P < 0.01$; $>$ or $<$, $P < 0.05$; NS, $P \geq 0.05$. (**a**–**c**) Ratios of female to male gene expression in several tissues of *D. pseudoobscura*. The 1,672 *D. pseudoobscura* expressed genes whose orthologues are present in *D. obscura* and have not experienced any inter-chromosomal translocation after splitting from *D. obscura* were analysed. Tissues analysed are as follows: (**a**) Ovaries in females and testes in males, (**b**) abdomens, and (**c**) abdomens after removing gonads. The lines in the boxes represent medians, 50% of values are included in the boxes, and 80% of values are included within the bars. $X_F$-$Y_F$, a group of genes whose orthologues are functional on both neo-X and neo-Y; $X_F$-$Y_P$, a group of genes whose orthologues are functional on the neo-X and non-functional on the neo-Y; $X_P$-$Y_F$, a group of genes whose orthologues are non-functional on the neo-X and functional on the neo-Y; $X_P$-$Y_P$, a group of genes whose orthologues are non-functional on both neo-X and neo-Y. The Mann-Whitney $U$ test also showed the same trend. (**d**) Ratio of the number of pseudogenes to the number of functional genes depending on tissues with maximum expression (cFPKM, see Supplementary Methods for details). Error bars indicate the 95% confidence interval based on a Monte Carlo simulation with 1,000 bootstrap replicates. Numbers in parentheses are the numbers of genes with maximum expression in the respective tissues. In this analysis, only genes for which a cFPKM in a tissue was at least twofold compared with that in any of other tissues examined were considered to remove the genes for which cFPKMs were similar in multiple tissues. See Supplementary Fig. 5 for other tissues and Supplementary Fig. 7 for the results under different conditions.

**Table 1 | Relationship between functionality and sexual antagonism for neo-X-linked genes in *Drosophila miranda*.**

| Functionality | Fb/Md* | Mb/Fd† | No conflict | Unclassified | Total |
|---|---|---|---|---|---|
| Functional | 29 | 51 | 865 | 4 | 949 |
| Pseudo | 16 | 13 | 287 | 17 | 333 |
| Total | 45 | 64 | 1,152 | 21 | 1,282 |

Sexually antagonistic genes in *D. miranda* were identified based on a homology search against *D. melanogaster* genes, among which Innocenti and Morrow[31] listed putative sexually antagonistic genes. The ratios of Fb/Md genes to Mb/Fd genes for functional genes and pseudogenes were different with marginal significance ($\chi^2$ test: $P = 0.076$).
*Genes with female-benefit/male-detriment effects.
†Genes with male-benefit/female-detriment effects.

If accelerated pseudogenization on the neo-X has mainly been caused by feminization as above, the rate of pseudogenization on the neo-X and the orthologous autosome in *D. pseudoobscura* may become comparable when the genes belonging to the $X_P$-$Y_F$ category were removed from the analysis. Indeed, when the number of pseudogenization events was counted under this condition, the difference in the number of pseudogenization events on the neo-X and the orthologous autosome in the *D. pseudoobscura* lineage (branches *X* and *Pse*, respectively, in Fig. 2b) became much smaller under the emergence times of the neo-sex chromosomes of 1 Mya (168 and 100.5 on the branches *X* and *Pse*, respectively, $\chi^2$ test: $\chi^2 = 17.0$, d.f. $= 1$, $P = 3.8 \times 10^{-5}$) and of 1.5 Mya (168 and 150.8, $\chi^2$ test: $\chi^2 = 0.9$, d.f. $= 1$, $P = 0.33$). Therefore, the acceleration of pseudogenization on the neo-X would mainly be caused by feminization of the neo-X.

**Effect of sexual antagonism on pseudogenization.** The above analysis clarified that sex-biased transmission affected the fates of neo-X-linked genes. Since the genes involved in sexual antagonism (that is, genes with female-benefit/male-detriment effects and genes with male-benefit/female-detriment effects) may also be affected by sex-biased transmission[19,30], we also examined the relationship between sexual antagonism and pseudogenization on the neo-X. As mentioned above, two-thirds of X chromosomes

are transmitted through females, which may result in larger effects of X-linked genes in females than in males. However, males are expected to be more sensitive to recessive genetic changes on the X chromosome. In this case, the effect of X-linked genes would be greater in males than in females. In other words, dominance is a key for the fate of sexually antagonistic genes on the X chromosome.

Unfortunately, sexually antagonistic genes in *D. miranda* have not been identified yet. However, candidates of sexually antagonistic genes have extensively been identified in *D. melanogaster*[31]. Assuming that sexually antagonistic genes have largely remained unchanged during *Drosophila* evolution, we examined the pattern of pseudogenization of sexually antagonistic genes on the neo-X in *D. miranda*. The pseudogenes on the neo-X showed a higher ratio of genes with female-benefit/male-detriment effects to genes with male-benefit/female-detriment effects than functional genes (16/13 [1.23] versus 29/51 [0.57], respectively), although the difference was only marginally significant ($\chi^2$ test: $\chi^2 = 3.1$, d.f. $= 1$, $P = 0.076$, Table 1). In other words, genes that are beneficial to females but deleterious to males tended to be pseudogenized on the neo-X. This result implies that many of the female-benefit/male-detriment alleles are recessive, rather than dominant. In this case, beneficial effects on females are mostly masked by other dominant alleles, particularly when allele frequency is low, while

deleterious effects on males are always exposed in the hemizygous condition of the neo-X. Therefore, the fitness effects of these genes in males on the neo-X may be higher than in females, resulting in the pseudogenization of such genes with female-benefit/male-detriment effects on the neo-X. However, the assumption that the sexually antagonistic genes in *D. miranda* are essentially the same as those in *D. melanogaster* may be unrealistic. Therefore, it is at this moment difficult to make any clear conclusions regarding the effects of sexual antagonism on pseudogenization of sex chromosomes. Further experimental as well as statistical studies are necessary to evaluate this hypothesis.

## Discussion

In this study, we revealed that the young X chromosome, that is, the neo-X, which emerged approximately 1 Mya in *D. miranda*, experienced accelerated pseudogenization compared with the pseudogenization taking place on autosomes. We also clearly showed that the neo-Y had an even higher rate of pseudogenization. Therefore, at least in the case of *D. miranda*, not only the neo-Y but also the neo-X rapidly degenerates in the initial stage of sex chromosome evolution. If this process of pseudogenization continues on both the neo-X and neo-Y, these neo-sex chromosomes would become evolutionary dead-ends. In reality, however, the number of functional genes on an X is roughly comparable to the numbers on other autosomes in many organisms[32,33]. This general trend suggests that the degeneration of an X would not continue for a long evolutionary time period; rather, the X maintains many functional genes and loses genes that are functionally unimportant or deleterious for one sex during the initial stage of sex chromosome evolution. One might think that the pseudogenization rate on the XR (or 2nd-X) in *D. miranda* and *D. pseudoobscura* can be examined in comparison with *D. obscura*, in which the orthologous chromosome is an autosome. Unfortunately, however, we have not yet had any appropriate outgroup species with high quality data of the genome, transcriptome and gene expression, which is critical in determining functionality of genes on the XR in our approach. (Note that *D. melanogaster* is not suitable for this purpose, because the three species were diverged from *D. melanogaster* more than 50 Mya[34]). Once such data become available, it will be interesting to examine the pseudogenization rate on the XR.

Sex chromosomes have been replaced many times during evolution[1,35], in which process an X, but not a Y, is expected to become a proto-autosome[36]. Yet, if the accelerated pseudogenization of an X is a general phenomenon, the reversal of the X to an autosome may occur via a more restricted route than previously thought. In other words, not only the Y but also the X seems to be highly deteriorated and specialized, making the reversal to an autosome more difficult. Of course, if the functions of pseudogenes on the neo-X are taken over by other genes (most likely duplicate genes) on other chromosomes, the sex chromosome reversal may still be possible with ease. Indeed, when we estimated the proportion of neo-X-linked genes that have duplicates on other chromosomes, pseudogenes on the neo-X showed a slightly higher proportion than functional genes on the neo-X, although the statistical significance of the difference was marginal (Supplementary Fig. 8). Alternatively, accelerated pseudogenization of an X may be a phenomenon restricted to particular taxonomic groups. Turnover of sex chromosomes in *Drosophila*[36] indeed seems to be a relatively slow process compared with the turnover in other organisms such as the medaka fish[37,38] and wrinkled frog[39]. However, as we have discussed, efficacy of natural selection on the X and autosomes in many *Drosophila* species may essentially be the same in a simple theory. In this case, the difference in pseudogenization rates between the X and autosomes is expected to be less visible in *Drosophila* species compared with other organisms in which meiotic recombination between homologous autosomes occur in both females and males. Therefore, the acceleration of pseudogenization on the X in comparison with autosomes can be more conspicuous in such organisms. In reality, however, other factors such as sexual selection should also be considered as we have discussed. Future studies focusing on a wide range of organisms with recent sex-chromosome reversals will provide further insights into the pseudogenization process of the X and the sex chromosome reversal during the long history of life.

In summary, we have clearly shown that the rate of pseudogenization was accelerated, not only on the neo-Y, but also on the neo-X, in *D. miranda*. Pseudogenization of the neo-X was likely caused by feminization due to the loss of male-biased genes and possibly by masculinization due to the loss of female-benefit/male-detriment genes. Genes under less functional constraint also tended to be pseudogenized on the neo-X. Therefore, the pseudogenization process at the initial stage of neo-X chromosome evolution involved a complex suite of evolutionary forces.

## Methods

**Flies.** *Drosophila miranda* (strain 14011-0101.17) and *D. obscura* (strain 14011-0151.01) were obtained from the *Drosophila* Species Stock Center at UC San Diego (https://stockcenter.ucsd.edu). *Drosophila pseudoobscura* with the known genome sequence [strain k-s12 (or 14011-0121.94)] was obtained from Kyorin-Fly (http://www.shigen.nig.ac.jp/fly/kyorin/cgi-bin/index.cgi).

**DNA extraction for library construction and sequencing.** We extracted total DNA from adult flies of *D. miranda* and *D. obscura*, using the method developed by Boom *et al.*[40]. Paired-end libraries (insert size of ∼300 bp) were constructed for *D. miranda*, whereas a paired-end library (insert size of ∼300 bp) as well as mate-pair libraries (insert size of ∼3 kbp and 8 kbp) were made for *D. obscura* by Macrogen (Seoul, South Korea). Paired-end sequencing of 101 bp for each of the libraries was performed by Macrogen with a HiSeq 2000 sequencer (Illumina, San Diego, CA).

**RNA extraction for library construction and sequencing.** Total RNA from different tissues was extracted using a standard acid phenol-guanidinium thiocyanate-chloroform extraction method[41] or using PureLink RNA Mini Kit (Thermo Fisher Scientific, Waltham, MA). The total RNA was treated with DNase I to digest genomic DNA, and then mRNA was purified from total RNA using the NEBNext Poly(A) mRNA Magnetic Isolation Module (NEB, Ipswich, MA). cDNA libraries were constructed from mRNA using the NEBNext Ultra Directional RNA Library Prep Kit for Illumina (NEB). Paired-end sequencing of 100 or 101 bp was performed by Beijing Genomics Institute (Beijing, China) or Macrogen, respectively, with a HiSeq 2000 sequencer.

**Genome assembly.** For *D. miranda*, all obtained reads were processed with Cutadapt version 1.6 (ref. 42) and SolexaQA++ v3.1 (ref. 43) to remove adapter sequences and low quality sequences, respectively. The selected reads were then mapped onto the reference MSH22 genome (DroMir2.2.fa) using bwa-0.7.8 (ref. 44). All variants (SNPs and indels) of our strains were called by a standard workflow of Picard-1.115 (http://broadinstitute.github.io/picard/) and GATK3.1.1 (ref. 45). Using our original scripts, we then replaced the reference genome sequence with the variants. The same procedures were then conducted to call variants on the neo-X. We obtained the neo-Y assembly by replacing the neo-X sequence with the male-specific variants. It should be noted that the ratio of the male to female coverages on the neo-X was similar to those on the autosomes (Supplementary Fig. 9). In addition, when female or male reads were mapped onto the *D. miranda* (strain 14011-0101.17) genome that we sequenced in this study, the frequency of variants on the neo-X of this genome was ∼20 times higher in males than in females mostly due to the mapping of the neo-Y-derived reads on the neo-X in males (Supplementary Fig. 10). (Note that the frequency of variants in males and females was similar on other chromosomes.) Although some male-specific variants on the neo-X in males may not be derived from the neo-Y but from the neo-X just by chance (that is, variants on the neo-X might be present in male samples but absent in female samples), the proportion of such variants was only minor (see Supplementary Methods and Supplementary Table 11 for estimation). These statistics indicate that our replacement strategy to obtain the neo-Y assembly must be appropriate and work with high accuracy.

For *D. obscura* (strain 14011-0151.01), we conducted a *de novo* genome assembly using ALLPATHS-LG[46] with default options. For *D. pseudoobscura*, we used the available genome assembly (dpse-all-chromosome-r3.1.fasta), which was downloaded from FlyBase (http://flybase.org/).

**Transcriptome assembly.** To assemble the transcriptome sequence for each of the three species, we used female and male RNA-seq data from whole body larvae, whole body pupae, adult heads, adult thoraxes and adult abdomens (Supplementary Tables 4–6). The RNA-seq data from each sample were first processed with Cutadapt version 1.6 and SolexaQA + + v3.1 as mentioned above. The filtered paired sequences were then mapped onto the genome assembly for each species obtained above using TopHat v2.0.11 (ref. 47) with default options. The transcriptome for each tissue was then constructed by Cufflinks v2.2.1 (ref. 48) with default options. Since the transcripts in different tissues are frequently overlapping, the transcriptomes for these tissues were finally merged using the 'cuffmerge' command included in Cufflinks.

**Gene annotation and classification.** We predicted open reading frames within transcripts using TransDecoder-2.0.1 (ref. 49). We also predicted the open reading frames *ab initio* in each genome using Augustus-3.0.3 (ref. 50) with the fly default option (--species = fly). The detailed statistics of the annotation for each species are shown in Supplementary Table 7.

All annotated genes were classified as follows. (1) If a gene was expressed, contained an initiation as well as a stop codon, and its CDS length was equal to or greater than 80% of the average of other members of an orthologous group, the gene was regarded as 'functional'. (2) If a gene was expressed, contained an initiation and a stop codon, and its CDS length was less than 80% of the average of other orthologous members, the gene was regarded as being 'disrupted'. (3) If a gene was not expressed and its CDS length was equal to or greater than 80% of the average of other orthologous members, the gene was being 'silenced'. (4) If a gene was not expressed and its CDS length was less than 80% of the average of other orthologous members, the gene was regarded as being 'silenced and disrupted'. (5) If a gene was expressed, lacked either one of an initiation or a stop codon, and its flanking 100 nucleotides did not contain any ambiguous site, the gene was being 'disrupted'. (6) If a gene was expressed in at least one of the tissues, lacked either one of an initiation or a stop codon, and flanking 100 nucleotides contained at least one ambiguous site (that is, N), the gene was regarded as being 'unclassified'.

**Assignment of pseudogenization events.** To compare the rate of pseudogenization in each evolutionary lineage, we analysed the orthologues that are likely functional in *D. obscura*, are present in at least one of *D. miranda* or *D. pseudoobscura*, and have not experienced any inter-chromosomal translocation in the lineages of *D. miranda* or *D. pseudoobscura* after splitting from *D. obscura*. (In other words, all these genes were likely functional in the common ancestor of the three species.) Under a parsimony framework, we assigned the pseudogenization events on each of the lineages after the divergence from *D. obscura*. For example, if an orthologous gene is functional in *D. pseudoobscura* and *D. obscura* but silenced (disrupted, or deleted) on the neo-X and neo-Y, the pseudogenization is assumed to have occurred in the ancestral lineage of *D. miranda* (branch *Anc* in Fig. 2b). Yet, if the cause of pseudogenization of the gene is different (for example, silenced on the neo-X whereas disrupted on the neo-Y), the pseudogenization was assigned to the both neo-X (branch *X*) and neo-Y (branch *Y*) lineages, independently, but not to the ancestral lineage.

More detailed procedures for our experiments and analyses are provided in the Supplementary Methods.

**Code availability.** All Perl scripts in this study are available from the corresponding author upon reasonable request.

**Data availability.** All sequence data generated in this study are available in the DDBJ Sequence Read Archive (DRA, http://trace.ddbj.nig.ac.jp/dra/index_e.html) with the accession numbers DRA004463, DRA004464 and DRA004465. All other data generated or analysed during this study are available from the corresponding author on reasonable request.

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

## Acknowledgements

We thank Chie Iwamoto and Miu Kubota for helping with our experiments and Masanori Toda for species identification. We are also grateful to Junichi Imoto, Yukako Katsura, Sonoko Kinjo, Norikazu Kitamura, Kaoru Matsumoto, Masatoshi Nei, Masa-aki Yoshida and Ikuko Yuyama for their comments on earlier versions of the manuscript. This work was supported by grants from the National Institute of Genetics and JSPS KAKENHI (Grant Numbers 25711023 and 15K14585) to M.N.

## Author contributions

M.N. designed the research. M.N., K.O. and M.F. conducted the experiments. M.N. analysed the data. M.N., K.I. and T.G. wrote the paper.

## Additional information

**Competing financial interests:** The authors declare no competing financial interests.

