## [Peer Review File · Nature Communications]

Reviewers' comments:

Reviewer #1 (Remarks to the Author):

The manuscript by Nozawa et al. describes a screen for pseudogene formation in a lineage that has recently evolved both a neo-Y and neo-X chromosome. They confirm earlier work that showed that the non-recombining neo-Y accumulated pseudogenes at an accelerated rate, but they also found a smaller, but significant, acceleration of the neo-X. This pattern for the neo-X is new, intriguing, and of interest to a wide diversity of biologists.

The authors next describe some interesting patterns: 1) Most neo-Y-linked pseudogenes were silenced via transcriptional inactivation while most neo-X-linked pseudogenes were silenced via structural defects (e.g., truncation of gene length or an internal stop codon), 2) dN/dS ratios indicated that pseudogenes located on the neo-X were under lower functional constraint compared those on the neo-Y, and 3) pseudogenes on the neo-X tended to have lower expression in males compared to other genes on the neo-X that were not silenced. All of these analyses seemed reasonable.

The authors next examined the influence of sexual antagonism on gene silencing on the neo-X. This analysis is based on the premise that sexual antagonism determined by Morrow's group in *D. melanogaster* can be used to assess sexual antagonism in the distantly related *D. miranda*. This blind extrapolation is speculative and completely unjustified, so unless the authors can actually measure sexual antagonism in *D. miranda*, this part of the paper needs to be removed.

Another concern is the authors' claim that the strength of selection of the neo-X is the same as that on the autosomes. They write "In this case, unlike other organisms, natural selection is expected to be four-thirds times more efficient on the X than on autosomes, which offsets a reduction in the effective population size of the X chromosomes by three-fourths compared with autosomes. This prediction suggests that the efficacy of natural selection on the X and autosomes is almost at the same extent in *Drosophila* species, which in turn expects no acceleration of pseudogenization on the X compared with autosomes." This argument is simplistically ad hoc without any justification based on explicit theory (i.e., no citations) and ignores the influence of sexual selection on the effective number of males and females. Much of the logic used throughout the manuscript relies on this unjustified assumption, so the statement "efficacy of natural selection on the X and autosomes is almost at the same" must be justified or the relevant parts of the manuscript must be rewritten.

MINOR POINTS:

line 86: Need references for divergence times between chromosomes and species.

line 172 (and similar statements elsewhere, e.g., line 140, 149, 256): The authors state "More specifically, if the ratio of females to males is one-to-one, two-thirds of X

chromosomes are inherited through females, while Y chromosomes are inherited only through males." The ratios of 2/3 and 1/3 (2/3 of X's are inherited from females and 1/3 from males) apply no matter what the sex ratio.

line 251: The authors state: "In other words, not only the Y but also the X seems to be highly deteriorated and specialized, making the reversal to an autosome more difficult." - This statement is not true when the sex-specific function of a neo-X pseudogene has been taken over elsewhere in the genome.

line 469: Do the "Anc" pseudogenes = those found on both the neo-X and neo-Y? If so, how is the case of independent loss of function on the X and Y accounted for?

Reviewer #2 (Remarks to the Author):

Genes on the Y (or W) chromosome are well known to undergo degeneration. Generally it is thought that the genes on the X degenerate much less. In this manuscript, the authors investigate if genes on the X chromosome also undergo substantial degeneration. They take advantage of the young sex chromosomes in *Drosophila miranda*. Specifically, they conducted genome sequencing, transcriptome sequencing, and comparisons with 2 closely related *Drosophila* species that do not have young sex chromosomes. They found that the X chromosomes do have surprisingly high levels of gene degeneration (premature protein termination or loss of gene expression) with about 1/3 of the genes pseudogenized. This rate was greater than that for the autosomes (and less than Y, as expected). Molecular evolution and bioinformatics analyses revealed that the genes that undergo degeneration have less functional constraint, that gene degeneration is feminizing (or demasculinizing) the X, and that sexual antagonism may also play a weak role. Their study suggests that it would be worthwhile to test if accelerated degeneration of neo-X's may be a general feature.

I found this study interesting and well done. Notably, this is the first thorough examination of X-chromosome degradation, and the result of accelerated degeneration on the neo-X is fascinating. Additionally, the manuscript was clearly presented.

Some minor comments:

1) L199-201. The result that genes with highest expression in testes were pseudogenized is similar to results from Kaiser and Bachtrog 2014 BMC Genomics.

2) L333 and onward: Could the authors comment on how a cutoff for "disrupted" was chosen to be 80% of the average length of the other orthologous members? Related, what is the distribution (% lengths) of the disrupted genes? What are the patterns with a more stringent cutoff, say 50%?

Minor editing:

1) L105 (and similar): Could probably delete "by the X^2 test"

Reviewer #3 (Remarks to the Author):

The study, entitled "Accelerated pseudogenization on the neo-X chromosome in *Drosophila miranda*", tries to answer a central question in evolutionary biology: does X chromosome degenerate faster than autosomes? Particularly, the study takes advantage of recently (one million years ago) evolved neo-X chromosome in *Drosophila miranda*, and compares the genes to the homologs in *D. pseudoobscura* and *D. obscura* to define pseudogenization events. Since the neo-X in *D. miranda* was recently evolved, it is still very similar to the neo-Y in sequence. This high sequence similarity poses a big challenge to assemble the neo-Y genome and genes on it because the neo-X sequencing reads can be wrongly incorporated into neo-Y gene models. The study has been careful to handle this problem, for example, assembling neo-X chromosome using female samples only and then constructing neo-Y by replacing the homologous sites with male-specific variants. The study concludes that the neo-X in *D. miranda* has faster pseudogenization rate than autosomes and that this can be caused by pseudogenization of male-biased genes, less-constrained genes, and sexually antagonistic genes. However, some critical information on the methods isn't clear, preventing a full assessment of the study. Also some results may have alternative explanations or need be extended to support the conclusions. These aspects are detailed below:

Major concerns:

1. Please clarify how the pseudogenization events are timed in Fig. 2B, particularly those events assigned to the branch neo-X and neo-Y. This is the most important result where the major conclusion is based on, so each type of assignment has to be clearly stated. The description in the supplement isn't clear how the timing is done. For example, why put a pseudogenization event in both neo-X and neo-Y branches if a gene is disrupted but not silenced on the neo-X? Don't the authors need examine the homologous gene on the neo-Y? The study also does not explain why the 'Anc' branch has lower pseudogenization event rate than the corresponding portion on the *D.pse* branch (6.6% vs 8.9%). Can this be caused by the timing method?

2. The definition of pseudogenization should be clearly stated in the text, ideally gives a diagram, because this is fundamental to the study and important to evaluate the results by readers. Putting the definition in supplement is insufficient. Also, will choosing different thresholds for the definition affect the conclusions, such as 60% of the average lengths?

3. Using the reads from female tissues only in assembling neo-X transcripts may introduce a bias against male-biased genes on the neo-X. Specifically, if a neo-X gene is mainly expressed in males, it would not be assembled as a neo-X gene because there are no or two few sequencing reads in female samples. If this is the case, the genuine neo-X gene may be wrongly classified as absence (pseudogenized?) in *D. miranda*. This reasoning is consistent with the observation that male-biased genes in *D. pseudoobscura* are more likely pseudogenized. This issue can be relieved if gene models are directly predicted from

genome sequences only, say, using Augustus.

4. It is unclear how many ORFs were predicted by either of the two methods: based on the transcripts assembled by cufflinks and by Augustus. And how they are combined? Did the Augustus method use the genome assembly only or incorporate extra information such as the mRNA alignments to the genome assembly?

5. Male-biased genes show higher pseudogenization rates on neo-X than those on the autosomes, and the study explains it as X-linkage relieves the constraint on the genes because the corresponding homolog on neo-Y can still serve the function. If this is true, one would expect that the neo-Y copy of a neo-X pseudogenized gene is functional. This is testable with the study's data. For example, after eliminating neo-X pseudogenes compensated by functional neo-Y homologs, will the pseudogenization rate of neo-X reduce to the level of an autosome?

6. The study uses the observation that in *D. miranda*, neo-X pseudogenized genes have higher dN/dS than functional genes (Fig. 3) to support functionally less important genes are more likely pseudogenized. However, the observation (higher dN/dS) in *D. miranda* can be a consequence, rather than cause, of pseudonization. The test need be done in a species representing ancestral status such as *D. pseudoobscura*, by comparing two gene groups there (both functional) having *D. miranda*'s functional and pseudogenized neo-X orthologs.

7. If that X-linkage increases pseudogenization rate is generally true in *Drosophila*, one expects to see that the XR in *D. pseudoobscura* and *D. miranda* also have, or previously experienced, higher pseudogenization rates? This can be tested by comparing to the homologous autosome in *D. obscura*.

8. Is there any reason for why the bootstrap/Monte-Carlo method is used for testing the difference of two groups given that standard method such as Mann-Whitney U Test is convenient and straightforward?

9. In the Introduction, the study is initially stated that theoretically X chromosome tends to degenerate because of smaller population size and lower recombination rates, and at the end states that this is not the case in *Drosophila* because of no meiotic recombination in males. This is confusing, so reorganize the Introduction, which had better focus on the case of studied species *Drosophila*.

Minor Concerns:

10. The study reasoned that X chromosome has larger effects in females than males because it is transmitted two thirds of the time. This is actually not true, because X chromosome is dosage compensated in males and as the study states later, the hemizygous X in males is more sensitive to recessive genetic changes. Based on this, the genetic changes overall may have larger effects in males than females. The authors may reconsider their writing to make the reasoning smoother.

11. Please clarify how the male and female SNPs in supplementary Fig. S5 are defined. Are

they the SNPs identified in the male and female DNA samples?

12. How many of the identified pseudogenized genes are fixed in population, e.g., present in both strains 14011-0101.17 and MSH22?

13. That assuming all male-specific variants are from neo-Y may overestimate the number of SNPs on neo-Y because by chance some neo-X SNPs may be present in male samples but missing in female ones. The proportion can be estimated by counting the SNPs on the XR of *D. miranda* that is present in male samples only. This issue needs be at least discussed.

14. Based on Fig. 2, the pseudogenization rate of the neo-X is about 2 fold that of the homologous *D. pseudoobscura*'s autosome, that is, one time higher, but the abstract states 'two times' higher.

15. The Abstract need state the result on the sexually antagonistic genes directly; it is vague when saying that the pseudogenization is biased for the sexually antagonistic genes on neo-X, as this may give the impression that the pseudogenization favors male beneficial genes based on the preceding statements, which is actually not the case.

Replies to all the comments from the reviewers:

Reviewer #1

1-1. The manuscript by Nozawa et al. describes a screen for pseudogene formation in a lineage that has recently evolved both a neo-Y and neo-X chromosome. They confirm earlier work that showed that the non-recombining neo-Y accumulated pseudogenes at an accelerated rate, but they also found a smaller, but significant, acceleration of the neo-X. This pattern for the neo-X is new, intriguing, and of interest to a wide diversity of biologists.

[Reply]

We are pleased to know that the reviewer recognizes the significance of our work.

1-2. The authors next describe some interesting patterns: 1) Most neo-Y-linked pseudogenes were silenced via transcriptional inactivation while most neo-X-linked pseudogenes were silenced via structural defects (e.g., truncation of gene length or an internal stop codon), 2) dN/dS ratios indicated that pseudogenes located on the neo-X were under lower functional constraint compared those on the neo-Y, and 3) pseudogenes on the neo-X tended to have lower expression in males compared to other genes on the neo-X that were not silenced. All of these analyses seemed reasonable.

[Reply]

We appreciate the reviewer to find our analyses reasonable.

1-3. The authors next examined the influence of sexual antagonism on gene silencing on the neo-X. This analysis is based on the premise that sexual antagonism determined by Morrow's group in *D. melanogaster* can be used to assess sexual antagonism in the distantly related *D. miranda*. This blind extrapolation is speculative and completely unjustified, so unless the authors can actually measure sexual antagonism in *D. miranda*, this part of the paper needs to be removed.

[Reply]

We agree with the reviewer that our analysis on sexual antagonism is speculative. Following the suggestion, we have further weakened our statement regarding this aspect (lines 31-33, lines 296-300, and lines 311-314). Please note that we now clearly discuss the limitation of our analysis. However, we still would

like to keep the section about the effect of sexual antagonism on pseudogenization of sex chromosomes, because we believe that examining such relationship with currently available data is meaningful and could stimulate further studies regarding this issue. We hope this revision is reasonable and understandable for the reviewer.

1-4. Another concern is the authors' claim that the strength of selection of the neo-X is the same as that on the autosomes. They write "In this case, unlike other organisms, natural selection is expected to be four-thirds times more efficient on the X than on autosomes, which offsets a reduction in the effective population size of the X chromosomes by three-fourths compared with autosomes. This prediction suggests that the efficacy of natural selection on the X and autosomes is almost at the same extent in *Drosophila* species, which in turn expects no acceleration of pseudogenization on the X compared with autosomes." This argument is simplistically ad hoc without any justification based on explicit theory (i.e., no citations) and ignores the influence of sexual selection on the effective number of males and females. Much of the logic used throughout the manuscript relies on this unjustified assumption, so the statement "efficacy of natural selection on the X and autosomes is almost at the same" must be justified or the relevant parts of the manuscript must be rewritten.

[Reply]

We appreciate the reviewer's comment. As the reviewer indicated, our assumption was too much simplified. Following the suggestion, we now mentioned about other factors such as the effective numbers of males and females with a reference (Singh and Petrov 2007). We also added two references (Vicoso and Charlesworth 2009; Charlesworth 2012) about theoretical as well as empirical aspects of effective population size and recombination rate of X and autosomes. These aspects are now clearly discussed in the main text (lines 167-186). We also clearly discussed about the effect of sexual selection on pseudogenization with references (Aguade 1999; Bono et al. 2015) (lines 209-212, see also lines 356-359).

1-5. line 86: Need references for divergence times between chromosomes and species.

[Reply]

We have added Bachtrog and Charlesworth (2002) as a reference (line 94).

1-6. line 172 (and similar statements elsewhere, e.g., line 140, 149, 256): The authors state "More specifically, if the ratio of females to males is one-to-one, two-thirds of X chromosomes are inherited through females, while Y chromosomes are inherited only through males." The ratios of 2/3 and 1/3 (2/3 of X's are inherited from females and 1/3 from males) apply no matter what the sex ratio.

[Reply]

We appreciate the reviewer's comment. To avoid any complication and confusion, we removed the words "if the ratio of females to males is one-to-one," as well as other relevant words from the main text (lines 222-223).

1-7. line 251: The authors state: "In other words, not only the Y but also the X seems to be highly deteriorated and specialized, making the reversal to an autosome more difficult." - This statement is not true when the sex-specific function of a neo-X pseudogene has been taken over elsewhere in the genome.

[Reply]

We greatly appreciate the comment. To examine the possibility that functions of neo-X-linked genes are taken over by other genes (in particular duplicate genes) on other chromosomes, we have searched the paralogous genes of all neo-X-linked genes in the *D. miranda* genome. The result showed that pseudogenes on the neo-X tend to have duplicates on other chromosomes with a bit higher frequency than functional genes on the neo-X, although statistical significance was marginal. Therefore, as the reviewer implies, reversal of sex chromosomes to autosomes may be possible with these duplicates on other chromosomes. The information is now provided as Supplementary Fig. 8 and clearly discussed in the main text (lines 342-347).

1-8. line 469: Do the "Anc" pseudogenes = those found on both the neo-X and neo-Y? If so, how is the case of independent loss of function on the X and Y accounted for?

[Reply]

We apologize that we did not include such detailed procedures in the main text but just put them into *Supplemental Experimental Procedures*. We now added the information to the Methods section (lines 450-461). In short, if the cause of pseudogenization of the gene is the same (e.g., silencing of the both neo-X and

neo-Y homologs), the pseudogenization was assigned to the ancestral lineage. If the cause of pseudogenization of the gene is different (e.g., silencing of the neo-X homolog whereas disruption of the neo-Y homolog), the pseudogenization was assigned to the both neo-X (branch X) and neo-Y (branch Y) lineages, independently, but not to the ancestral lineage.

Reviewer #2

2-1. Genes on the Y (or W) chromosome are well known to undergo degeneration. Generally it is thought that the genes on the X degenerate much less. In this manuscript, the authors investigate if genes on the X chromosome also undergo substantial degeneration. They take advantage of the young sex chromosomes in *Drosophila miranda*. Specifically, they conducted genome sequencing, transcriptome sequencing, and comparisons with 2 closely related *Drosophila* species that do not have young sex chromosomes. They found that the X chromosomes do have surprisingly high levels of gene degeneration (premature protein termination or loss of gene expression) with about 1/3 of the genes pseudogenized. This rate was greater than that for the autosomes (and less than Y, as expected). Molecular evolution and bioinformatics analyses revealed that the genes that undergo degeneration have less functional constraint, that gene degeneration is feminizing (or demasculinizing) the X, and that sexual antagonism may also play a weak role. Their study suggests that it would be worthwhile to test if accelerated degeneration of neo-X's may be a general feature.

[Reply]

We are very glad to know that the reviewer nicely summarized our work. This is exactly what we would like to emphasize.

2-2. I found this study interesting and well done. Notably, this is the first thorough examination of X-chromosome degradation, and the result of accelerated degeneration on the neo-X is fascinating. Additionally, the manuscript was clearly presented.

[Reply]

We are very glad to know that the reviewer recognizes the importance of our work.

2-3. L199-201. The result that genes with highest expression in testes were

pseudogenized is similar to results from Kaiser and Bachtrog 2014 BMC Genomics.

[Reply]

We thank the reviewer for providing us with the related work. We now clearly state that the results are consistent with those of Kaiser and Bachtrog (2014) (line 253). In relation to this point, Kaiser and Bachtrog (2014) mentioned that neo-sex chromosomes in *D. miranda* emerged 1.5 Mya, which was older than the time that we originally used in this study (i.e., 1 Mya). Since this difference may affect our conclusion that pseudogenization is accelerated on the neo-X, we have analyzed the rate of pseudogenization based on this older emergence time as well. The result showed that the rate of pseudogenization was still significantly faster for the neo-X than the orthologous autosome in *D. pseudoobscura*. These analyses are now clearly stated in the main text (lines 108-113) and presented as Supplementary Figure 2. We believe that this additional analysis has made our conclusion more robust.

2-4. L333 and onward: Could the authors comment on how a cutoff for "disrupted" was chosen to be 80% of the average length of the other orthologous members? Related, what is the distribution (% lengths) of the disrupted genes? What are the patterns with a more stringent cutoff, say 50%?

[Reply]

Following the comment, we examined the rate of pseudogenization with the different thresholds of CDS length from 10 to 90% with a 10%-interval. The results are now provided in Supplementary Tables 8 and 9. The results consistently supported the higher pseudogenization rate on the neo-X in *D. miranda* compared with the rate on the orthologous autosome in *D. pseudoobscura*. This is now clearly stated in the main text (lines 113-121). Therefore, we believe that our conclusion has become more robust.

Regarding the distribution of relative CDS length of the genes on the neo-X and the neo-Y, we now provide the diagram below just for the reviewing process. However, to avoid any more complication, we decided not to add this figure to the main text nor supplementary material. We hope this is understandable for the reviewer.

2-5. L105 (and similar): Could probably delete "by the χ^2 test"

[Reply]

Following this suggestion, we deleted "by the χ^2 test" from the main text. However, we have kept the words in Table 1 and Figure 2 for readability.

Reviewer #3

3-0. The study, entitled "Accelerated pseudogenization on the neo-X chromosome in

Drosophila miranda", tries to answer a central question in evolutionary biology: does X chromosome degenerate faster than autosomes? Particularly, the study takes advantage of recently (one million years ago) evolved neo-X chromosome in *Drosophila miranda*, and compares the genes to the homologs in *D. pseudoobscura* and *D. obscura* to define pseudogenization events. Since the neo-X in *D. miranda* was recently evolved, it is still very similar to the neo-Y in sequence. This high sequence similarity poses a big challenge to assemble the neo-Y genome and genes on it because the neo-X sequencing reads can be wrongly incorporated into neo-Y gene models. The study has been careful to handle this problem, for example, assembling neo-X chromosome using female samples only and then constructing neo-Y by replacing the homologous sites with male-specific variants. The study concludes that the neo-X in *D. miranda* has faster pseudogenization rate than autosomes and that this can be caused by pseudogenization of male-biased genes, less-constrained genes, and sexually antagonistic genes. However, some critical information on the methods isn't clear, preventing a full assessment of the study. Also some results may have alternative explanations or need be extended to support the conclusions. These aspects are detailed below:

[Reply]

Following all the comments, we have seriously and extensively revised the manuscript. In the followings, we replied to all of the comments one by one.

3-1. Please clarify how the pseudogenization events are timed in Fig. 2B, particularly those events assigned to the branch neo-X and neo-Y. This is the most important result where the major conclusion is based on, so each type of assignment has to be clearly stated. The description in the supplement isn't clear how the timing is done. For example, why put a pseudogenization event in both neo-X and neo-Y branches if a gene is disrupted but not silenced on the neo-X? Don't the authors need examine the homologous gene on the neo-Y? The study also does not explain why the 'Anc' branch has lower pseudogenization event rate than the corresponding portion on the D.pse branch (6.6% vs 8.9%). Can this be caused by the timing method?

[Reply]

As we have written in the main text, we assigned each of the pseudogenization events under a parsimony framework with a constraint that each of the genes was functional in the ancestor of these three species. Therefore, if an orthologous gene is functional in *D. obscura* and *D. pseudoobscura* but

pseudogenized in the both neo-X and neo-Y in *D. miranda*, we regarded that the ortholog became pseudogenes in the *D. miranda* lineage before the emergence of neo-sex chromosomes. However, independent pseudogenization events in both lineages after splitting into neo-X and neo-Y are also possible. To take account of such a possibility, when a possible causal mutation for pseudogenization is different between the homologous genes on the neo-X and neo-Y, we regarded that the pseudogenization occurred independently in both lineages after becoming neo-sex chromosomes. For example, if the gene on the neo-X is silenced, while the homolog on the neo-Y is disrupted, we regarded that the pseudogenization occurred independently in both neo-X and neo-Y lineages. In other words, we did examine the homologous genes on the neo-sex chromosomes. This explanation was now added to the Methods section (lines 450-461).

Regarding the second point about a smaller number (84) of pseudogenization events on the branch *Anc* compared with the number (114.5) on the branch *Pse*, the emergence time of the neo-sex chromosomes affects the number of pseudogenization events on the counterpart of the branch *Pse*, as the reviewer indicated. Indeed, when 1.5 Mya was used as the emergence time of the neo-sex chromosomes, the relative number (84) of pseudogenization events on the branch *Anc* became greater than the number (57.2) on the counterpart of the branch *Pse*. Therefore, the actual emergence time of the neo-sex chromosomes might be between these two time points (i.e., neither 1 nor 1.5 Mya but between them) in which the number of pseudogenization events on the two branches is similar, although this argument may be too speculative at this moment. Please note that even in this case, the pseudogenization rate was significantly higher for the neo-X lineage (branch *X*) than the *D. pseudoobscura* lineage (branch *Pse*). The above information is now added to the main text (lines 108-113) as well as provided as Supplementary Figure 2.

3-2. The definition of pseudogenization should be clearly stated in the text, ideally gives a diagram, because this is fundamental to the study and important to evaluate the results by readers. Putting the definition in supplement is insufficient. Also, will choosing different thresholds for the definition affect the conclusions, such as 60% of the average lengths?

[Reply]

We have clearly stated the definition of pseudogenes in the main text (lines

430-447) as well as in *Supplemental Experimental Procedures*.) Following the comment, we additionally examined the rate of pseudogenization with the different thresholds from 10 to 90% with a 10%-interval. The results are now provided in Supplementary Tables 8 and 9. The results consistently supported the higher pseudogenization rate on the neo-X in *D. miranda* compared with the rate on the orthologous autosome in *D. pseudoobscura*. This is now clearly stated in the main text (lines 113-121). Our conclusion became more robust, thanks to the reviewer.

3-3. Using the reads from female tissues only in assembling neo-X transcripts may introduce a bias against male-biased genes on the neo-X. Specifically, if a neo-X gene is mainly expressed in males, it would not be assembled as a neo-X gene because there are no or two few sequencing reads in female samples. If this is the case, the genuine neo-X gene may be wrongly classified as absence (pseudogenized?) in *D. miranda*. This reasoning is consistent with the observation that male-biased genes in *D. pseudoobscura* are more likely pseudogenized. This issue can be relieved if gene models are directly predicted from genome sequences only, say, using Augustus.

[Reply]

Following this suggestion from the reviewer, we have examined the expression level of genes predicted by Augustus. Please note that in this case the genes solely predicted by Augustus can also be regarded as functional genes if they are expressed in at least one tissue. The results obtained showed essentially the same trend as those based on the annotation by Cufflinks/TransDecoder only, but the difference as well as the statistical significance among categories became smaller. Therefore, we admit that our approach was certainly biased to some extent, though it was almost inevitable to clearly separate the reads from the neo-X and the neo-Y. Yet, please note that, the genes whose orthologs are pseudogenized on the neo-X remained to be male biased in many tissues of *D. pseudoobscura* in this new analysis as well. Therefore, the pattern we originally found is not an artifact due to our approach of gene annotation but authentic. These results are now provided as Supplementary Fig. 7, and clearly stated in the main text (lines 256-272).

This insightful comment raised another concern about the acceleration of pseudogenization on the neo-X. If our annotation of neo-X-linked genes was biased as the reviewer pointed out, it may cause overestimation of the number of silenced genes on the neo-X lineage. This is quite critical for our main conclusion. Therefore,

we also examined the number of pseudogenization events on each of the lineages by considering the expression level of genes that were annotated either by Cufflinks/TransDecoder or Augustus. The results showed essentially the same trend. More specifically, the number of pseudogenization events on the neo-X lineage was still significantly greater than the number on the corresponding autosomal lineage in *D. pseudoobscura*. Therefore, our conclusion that the neo-X is under an accelerated pseudogenization after becoming sex chromosomes was robust irrespective of the approaches of gene annotation. This result is now provided as Supplementary Fig. 3 and described in the main text (lines 122-139).

In relation to this argument, when we considered the expression level of the genes predicted by Augustus, many genes were judged as expressed if the cutoff FPKM was set to 1 in at least one of the tissues. In other words, the number of silenced genes became much smaller than the number based only on TransDecoder. Of course, the above criterion for expressed genes may be too loose and apparently underestimate the number of silenced pseudogenes. It is therefore at this moment very difficult to make any decisive conclusion regarding this aspect. Further analyses as well as experiments are apparently necessary. This information is now provided as Supplementary Fig. 4 and described in the main text (lines 149-165). We again appreciate the reviewer for his/her insightful comment. We believe that our work now becomes more solid and reliable.

3-4. It is unclear how many ORFs were predicted by either of the two methods: based on the transcripts assembled by cufflinks and by Augustus. And how they are combined? Did the Augustus method use the genome assembly only or incorporate extra information such as the mRNA alignments to the genome assembly?

[Reply]

For gene prediction by Augustus, we only used a genome sequence without any other data (just with a “fly” option). When TransDecoder and Augustus predicted ORFs in an overlapping region on a genome, the annotation based on TransDecoder was preferentially adopted because TransDecoder is based on the transcriptome data so that it is clear that the region is at least transcribed. This statement is now added to the Supplementary Experimental Procedures, section “Gene annotation”. In addition, the numbers of genes predicted by TransDecoder (+Cufflinks) and Augustus are now added to Supplementary Table 7.

3-5. Male-biased genes show higher pseudogenization rates on neo-X than those on the autosomes, and the study explains it as X-linkage relieves the constraint on the genes because the corresponding homolog on neo-Y can still serve the function. If this is true, one would expect that the neo-Y copy of a neo-X pseudonized gene is functional. This is testable with the study's data. For example, after eliminating neo-X pseudogenes compensated by functional neo-Y homologs, will the pseudogenization rate of neo-X reduce to the level of an autosome?

[Reply]

We originally classified all neo-sex-linked genes into four categories (i.e., functional on both chromosomes, functional only on the neo-X, functional only on the neo-Y, and nonfunctional on both chromosomes). To enhance the readability of our manuscript, however, before submission we decided to classify the genes based only on the functionality of the neo-X-linked genes without considering the functionality of the neo-Y-linked genes. Because the reviewer kindly made this point, we now put our original idea of four gene categories back to the manuscript. As we mentioned above, we divided all genes on the neo-sex-linked genes into four categories, i.e., X_F-Y_F , X_F-Y_P , X_P-Y_F , and X_P-Y_P , respectively. Based on this classification of genes, we have analyzed the d_N/d_S and the F/M ratio of gene expression. The results obtained showed that X_P-Y_F genes evolve significantly faster than other categories of genes in the ancestor before becoming the neo-sex chromosomes (Figure 3). Moreover, those genes are male-biased (Figure 4, see also Supplementary Figs. 5 and 7). These results are consistent because some genes expressed in testes and accessory glands are known to evolve faster under sexual selection. Since these genes are unlikely to be important for females, they would have been pseudogenized after the chromosome became the neo-sex chromosomes. In addition, when these X_P-Y_F genes were removed from the analysis, the difference in pseudogenization rates between the neo-X and its orthologous autosome in *D. pseudoobscura* became much less significant, being consistent with our idea that the pseudogenization on the neo-X has been accelerated due to feminization/demasculinization. These analyses and discussions are now clearly mentioned in the main text (lines 193-218, 231-249, and 274-283). These additional analyses have made our conclusion more solid.

3-6. The study uses the observation that in *D. miranda*, neo-X pseudogenized genes have higher d_N/d_S than functional genes (Fig. 3) to support functionally less important

genes are more likely pseudogenized. However, the observation (higher d_N/d_S) in *D. miranda* can be a consequence, rather than cause, of pseudonization. The test need be done in a species representing ancestral status such as *D. pseudoobscura*, by comparing two gene groups there (both functional) having *D. miranda*'s functional and pseudogenized neo-X orthologs.

[Reply]

We conducted the analyses to infer the functional importance of the ancestral genes before becoming the neo-sex chromosomes. More specifically, we have estimated the d_N/d_S ratio on the ancestral branch (*Anc*) in Figure 2B. Analyzing this branch would distinguish whether the higher d_N/d_S ratio for the genes that are pseudogenized on the neo-X is a consequence after pseudogenization. The results showed that the d_N/d_S ratio for X_P - Y_F genes is higher than those for other categories of genes not only on the branch *X* but also on the branch *Anc*. Therefore, the high d_N/d_S ratio of X_P - Y_F genes is unlikely to be a consequence after pseudogenization. Rather, this indicates that less important genes tended to become pseudogenes on the neo-X after becoming the neo-sex chromosomes. X_P - Y_P genes also showed a similar trend but less conspicuous. This is now clearly stated in the main text (lines 193-218) as well as shown in Figure 3.

3-7. If that X-linkage increases pseudogenization rate is generally true in *Drosophila*, one expects to see that the XR in *D. pseudoobscura* and *D. miranda* also have, or previously experienced, higher pseudogenization rates? This can be tested by comparing to the homologous autosome in *D. obscura*.

[Reply]

We agree with the reviewer's comment. We have also been interested in very much to estimate the pseudogenization rate on the XR in *D. pseudoobscura* and *D. miranda*. Unfortunately, however, we need one more outgroup species to analyze the XR, because *D. obscura* in this case must be a comparing species rather than outgroup species. This outgroup species is particularly important to determine the functionality of each gene. At this moment, we do not have any such species with comparable data of genomes, transcriptomes, and gene expression. Please note that *D. melanogaster* and these three species diverged more than 50 Mya (Tamura et al. 2004), which makes the analysis less reliable. Therefore, we refrain, at this moment, conducting such an analysis. Once such outgroup species

has become available, we are willing to estimate the pseudogenization rate on the XR. This explanation is now provided in the main text (lines 329-336).

3-8. Is there any reason for why the bootstrap/Monte-Carlo method is used for testing the difference of two groups given that standard method such as Mann-Whitney U Test is convenient and straightforward?

[Reply]

We originally used the Mann-Whitney U test as well. The statistical trend of the bootstrap test was the same as the Mann-Whitney U test, but the Mann-Whitney U test gave stronger statistical significance than the bootstrap test. As an example, we showed the statistical significance based on the bootstrap test and the Mann-Whitney U test for the data shown in Fig. 3. To make our analyses conservative, however, we decided to use the bootstrap test. This information is now provided in the figure legends (line 630 and lines 639-640). Please note that to avoid any further complication we decided not to add the results based on the Mann-Whitney U test to our figures. We hope this explanation is understandable to the reviewer.

3-9. In the Introduction, the study is initially stated that theoretically X chromosome tends to degenerate because of smaller population size and lower recombination rates, and at the end states that this is not the case in *Drosophila* because of no meiotic recombination in males. This is confusing, so reorganize the Introduction, which had better focus on the case of studied species *Drosophila*.

[Reply]

Following the suggestion from the reviewer, we removed the statements about effective population size and recombination rate of X and autosomes from the Introduction section. We also reorganized and shortened the section (lines 48-50).

3-10. The study reasoned that X chromosome has larger effects in females than males because it is transmitted two thirds of the time. This is actually not true, because X chromosome is dosage compensated in males and as the study states later, the hemizygous X in males is more sensitive to recessive genetic changes. Based on this, the genetic changes overall may have larger effects in males than females. The authors may reconsider their writing to make the reasoning smoother.

[Reply]

Following the comment, we now clearly state the high sensitivity to recessive genetic changes to males (lines 291-295). We also understand that effect of dosage compensation on the evolution of the male X chromosome. Actually, we have also worked on the evolution of dosage compensation in *Drosophila* species (Nozawa et al. 2014 MBE). However, to avoid any further complication, we decided not to mention about it in this work. We hope our way of revision is understandable.

3-11. Please clarify how the male and female SNPs in supplementary Fig. S5 are defined. Are they the SNPs identified in the male and female DNA samples?

[Reply]

In the present study, male (or female) variants (SNPs + indels) mean that male (or female) reads contained different nucleotide(s) compared with the obtained *D. miranda* genome (strain 14011-0101.17) at a nucleotide position when the reads were mapped onto the genome. This statement is now provided in the

Methods section (lines 402-413) as well as in Supplementary Experimental Procedures.

3-12. How many of the identified pseudogenized genes are fixed in population, e.g., present in both strains 14011-0101.17 and MSH22?

[Reply]

We have also been interested in estimating the number of segregating pseudogenes (i.e., functional in some individuals/populations but nonfunctional in others). However, since the ways of gene annotation are different between us (strain 14011-0101.17) and Bachtrog's group (strain MSH22), it would not be reasonable to compare these data. (Please also note that the tissues for RNA-seq for constructing the transcriptome of *D. miranda* are also different between us and her group.) We are now sequencing the genome and transcriptome of another strain of *D. miranda* with the same strategy that we have adopted in this study. Once it is completed, it will be interesting to see such aspects.

3-13. That assuming all male-specific variants are from neo-Y may overestimate the number of SNPs on neo-Y because by chance some neo-X SNPs may be present in male samples but missing in female ones. The proportion can be estimated by counting the SNPs on the XR of *D. miranda* that is present in male samples only. This issue needs be at least discussed.

[Reply]

Following this comment, we estimated the proportion of male-specific variants that were derived from the neo-X using the ratio of female- to male-specific variants on XL and XR as a control. Then, the proportion was only 1.3%. Therefore, the majority of the male-specific variants on the neo-X were likely to be truly derived from the neo-Y. The information is now provided as Supplementary Table 11 and described in Supplementary Experimental Procedures. Moreover, we now discuss the possibility of overestimation of the neo-Y-derived SNPs in the main text (lines 402-413).

3-14. Based on Fig. 2, the pseudogenization rate of the neo-X is about 2 fold that of the homologous *D. pseudoobscura*'s autosome, that is, one time higher, but the abstract states 'two times' higher.

[Reply]

Following the comment, we removed “two times higher” from the manuscript. Instead, we now consistently use “2-fold” (lines 27-28 and lines 97 and 103).

3-15. The Abstract need state the result on the sexually antagonistic genes directly; it is vague when saying that the pseudogenization is biased for the sexually antagonistic genes on neo-X, as this may give the impression that the pseudogenization favors male beneficial genes based on the preceding statements, which is actually not the case.

[Reply]

Following the comment, we directly mentioned the analysis of sexually antagonistic genes in Abstract (lines 31-33). At the same time, however, we have weakened our statements about sexually antagonistic genes, because the analysis relies on a couple of assumptions that have not really been tested yet, as the reviewer #1 pointed out (see comment 1-3).

REVIEWERS' COMMENTS:

Reviewer #1 (Remarks to the Author):

After reviewing the changes made in the revised manuscript in response to my review, I can now recommend publication.

Reviewer #3 (Remarks to the Author):

I appreciate that the authors have made significant improvements by incorporating the reviewers' comments/suggestions and by adding new analyses. In particular, the manuscript has clarified the method to time pseudogenization events, used different CDS length thresholds to define pseudogenization events, considered the potential artifacts caused by the method of classifying silenced genes, and so on. The analyses in the revised version are more reliable than the previous one. The writing is also clearer by reorganizing the Introduction.

I however still have the following few comments, some are to restate my previous comments, and some are for the new analyses.

1. It is clear that the difference of pseudogenization rates between the neo-X in *D. miranda* and its homologous autosome in *D. pseudoobscura* becomes much smaller after controlling some confounding factors. For example, in Fig. S3B, the pseudogenization rates are 16.5% and 12.2% (i.e., ~1.3 fold), respectively, after removing the potential artifact caused by male-biased neo-X genes and using the 1.5My divergence time. The Abstract may weaken their relevant statement to reflect this uncertainty.
2. When different CDS length thresholds are used for defining 'disrupted' genes, the variation of the number of pseudogenes is surprisingly small (Table S8). For example, the thresholds 10% and 80% give 217 and 242 genes, respectively. This means that most of the pseudogenized genes have very short CDS length compared to their orthologs. For readers to reproduce/evaluate the results, the predicted/assembled gene models should be submitted as supplement in a GFF (or equivalent) format, in addition to the genome assemblies which should be deposited to a public database such as DDBJ.
3. Related to my previous comment 3-7: the authors argued no outgroup species to test whether the chromosome XR also show faster pseudogenization, but given the *D. obscura* species just sequenced by this study, it should be easy to compare the XR chromosome (from *D. pseudoobscura* or *D. miranda*) to the corresponding homologous autosome in *D. obscura* for a conclusion.
4. Related to my previous comment 3-6. It is unclear how synonymous and nonsynonymous substitution rates and their ratios are estimated based on the method description in the supplementary materials on Page 33. Were the orthologous sequences from *D. obscura*, *D. pseudoobscura*, and *D. miranda* all used in an alignment or just a subset of them? If the rate for the branch to *D. pseudoobscura* was estimated, showing it would be more effective than showing the 'Anc' branch to indicate whether the pseudogenized genes in *D. miranda* were under weaker constraints before becoming pseudogenized.

Minor:

1. In the Abstract, "... a weak trend that the genes with male-benefit/female detriment effects identified in *D. melanogaster* are pseudogenized on the neo-X", but in the text (Line 300-302), it shows that the genes with female-benefit/male detriment effects have such tendency.

Replies to all the comments from the reviewers:

Reviewer #1

1-1. After reviewing the changes made in the revised manuscript in response to my review, I can now recommend publication.

[Reply]

We are pleased to know that the reviewer is satisfied with our revisions.

Reviewer #3

3-1. I appreciate that the authors have made significant improvements by incorporating the reviewers' comments/suggestions and by adding new analyses. In particular, the manuscript has clarified the method to time pseudogenization events, used different CDS length thresholds to define pseudogenization events, considered the potential artifacts caused by the method of classifying silenced genes, and so on. The analyses in the revised version are more reliable than the previous one. The writing is also clearer by reorganizing the Introduction.

[Reply]

We are pleased to know that the reviewer is generally satisfied with our revisions.

3-2. I however still have the following few comments, some are to restate my previous comments, and some are for the new analyses.

[Reply]

Following the comments, we have again revised the manuscript. In the following, we replied all the comments from the reviewer one by one.

3-3. It is clear that the difference of pseudogenization rates between the neo-X in *D. miranda* and its homologous autosome in *D. pseudoobscura* becomes much smaller after controlling some confounding factors. For example, in Fig. S3B, the pseudogenization rates are 16.5% and 12.2% (i.e, ~1.3 fold), respectively, after removing the potential artifact caused by male-biased neo-X genes and using the 1.5My divergence time. The Abstract may weaken their relevant statement to reflect this uncertainty.

[Reply]

Following the comment, we have weakened our statement regarding this aspect in the Abstract (lines 24-26).

3-4. When different CDS length thresholds are used for defining ‘disrupted’ genes, the variation of the number of pseudogenes is surprisingly small (Table S8). For example, the thresholds 10% and 80% give 217 and 242 genes, respectively. This means that most of the pseudogenized genes have very short CDS length compared to their orthologs. For readers to reproduce/evaluate the results, the predicted/assembled gene models should be submitted as supplement in a GFF (or equivalent) format, in addition to the genome assemblies which should be deposited to a public database such as DDBJ.

[Reply]

We are pleased to provide a GFF file as well as genome assemblies upon requests. This is now clearly mentioned in the Data availability section (lines 468-472).

3-5. Related to my previous comment 3-7: the authors argued no outgroup species to test whether the chromosome XR also show faster pseudogenization, but given the *D. obscura* species just sequenced by this study, it should be easy to compare the XR chromosome (from *D. pseudoobscura* or *D. miranda*) to the corresponding homologous autosome in *D. obscura* for a conclusion.

[Reply]

We completely agree with the reviewer’s requirement. However, as we have mentioned previously, we are hesitant to analyze the XR without outgroup species. We strongly hope that our decision is acceptable for the reviewer.

3-6. Related to my previous comment 3-6. It is unclear how synonymous and nonsynonymous substitution rates and their ratios are estimated based on the method description in the supplementary materials on Page 33. Were the orthologous sequences from *D. obscura*, *D. pseudoobscura*, and *D. miranda* all used in an alignment or just a subset of them? If the rate for the branch to *D. pseudoobscura* was estimated, showing it would be more effective than showing the ‘Anc’ branch to indicate whether the pseudogenized genes in *D. miranda* were under weaker constraints before becoming

pseudogenized.

[Reply]

We now clearly mentioned that we analyzed the 1,282 orthologs which are likely to be functional in *D. obscura* and not to have experienced any inter-chromosomal translocation after the species split from *D. obscura* (page 34, lines 1-4 in the Supplementary Material). Therefore, the rates for synonymous and nonsynonymous sites were also estimated for the branch leading to *D. pseudoobscura*. To be honest, however, we cannot understand why showing the rates for the *D. pseudoobscura* branch (*Pse*) is more effective than showing the rates for the ancestral branch (*Anc*) before splitting the neo-X and neo-Y leading to the *D. miranda*. The branch *Anc* is a lineage that is a direct ancestor to the neo-X and neo-Y lineages, whereas the branch *Pse* is not a direct ancestral lineage of the neo-X and neo-Y but a different lineage strictly speaking. Therefore, showing the rates on the *Anc* lineage is apparently more effective for our analysis. For this reason, we decided to keep the current version of our analysis.

3-7. In the Abstract, "... a weak trend that the genes with male-benefit/female detriment effects identified in *D. melanogaster* are pseudogenized on the neo-X", but in the text (Line 300-302), it shows that the genes with female-benefit/male detriment effects have such tendency.

[Reply]

We deeply apologize for such a careless mistake. Our statement in the Abstract was wrong. We have corrected our Abstract (line 29). We appreciate the reviewer for his/her careful review.